# An Adaptive Roller Speed Control Method Based on Monitoring Value of Real-Time Seed Flow Rate for Flute-Roller Type Seed-Metering Device

**DOI:** 10.3390/s21010080

**Published:** 2020-12-25

**Authors:** Wei Liu, Jianping Hu, Xingsheng Zhao, Mengjiao Yao, Imran Ali Lakhiar, Jun Zhao, Jiaxin Liu, Wei Wang

**Affiliations:** School of Agricultural Engineering, Jiangsu University, Zhenjiang 212013, China; 2111616005@stmail.ujs.edu.cn (W.L.); 2221816022@stmail.ujs.edu.cn (X.Z.); 2221916042@stmail.ujs.edu.cn (M.Y.); 5103160321@stmail.ujs.edu.cn (I.A.L.); 2221816039@stmail.ujs.edu.cn (J.Z.); 2211916030@stmail.ujs.edu.cn (J.L.); 2111816011@stmail.ujs.edu.cn (W.W.)

**Keywords:** seeding rate, rotational speed, control system, expert control, seeding performance

## Abstract

In order to obtain desirable crop yields, grain seeds need to be sown at the optimal seed amount per hectare with uniform distribution in the field. In previous grain sowing processes, the seeding rates are controlled by the rotational speed of the flute roller which significantly effects the uniform distribution of the seeds due to disturbances, such as the reduction of the seeds’ mass in the hopper and the change of working length of the flute roller. In order to overcome the above problem, we developed an adaptive roller speed control system based on the seed flow rate sensor. The developed system can monitor and feedback actual seeding rates. In addition, based on the monitoring value of the real-time seeding rates, an adaptive roller speed control method (ARSCM), which contains an algorithm for calculating the seeding rate with a compensation, was proposed. Besides, the seeding performance of the ARSCM and that of the conventional roller speed control method (CRSCM) were compared. The results of constant-velocity experiments demonstrated that the accuracy (SA) and the coefficient of variation (SCV) of the seeding rates controlled by the ARSCM were 94.12% and 6.77%, respectively. As for the CRSCM, the SA and SCV were 89.00% and 8.95%, respectively. Under variable-velocity conditions, the SA and SCV of the proposed system were 91.58% and 11.08%, respectively, while those of the CRSCM were 88.48% and 13.08%, respectively. Based on the above results, this study concluded that the ARSCM is able to replace the CRSCM in practical sowing processes for the optimal and uniform seed distribution in the field.

## 1. Introduction

In order to obtain desirable crop yields, grain seeds not only need to be sown at the optimal seeding amount per hectare, but also should be distributed uniformly in the field [1,2,3]. Studies reported that to achieve the above aim, the seeding rates should be controlled as accurately as possible in a short distance. The grain seeds need to be sown in rows at a large seeding amount per hectare (54–269 kg ha^−1^) [4]. Compared with the disk-cell type and pneumatic type seed-metering devices [5,6], the flute-roller type seed-metering device can discharge the largest amount of the seeds in a rotation. Therefore, in order to meet the requirement of seeding efficiency of the grain seeds, the flute roller type seed-metering device was chosen to discharge the grain seeds [7].

The conventional grain drill controls the seeding rates by the mechanical transmission system which maintains a fixed transmission ratio between the ground wheel and the flute roller. Due to the slipping and spinning of the ground wheel, the mechanical transmission system may result in reseeding or miss-seeding phenomena, which significantly reduce the seeding accuracy and seeding uniformity [8]. In addition, some researchers developed electric seeding control systems and replaced the chain-sprockets or gears system for driving flute rollers with the electric motor [9]. Their studies reported that the performance of the grain drills with the electric driving systems was desirable, because the error of the seeding rate in the holistic sown field was 4.92%. The off-the-shelf electronic seeding control systems can be categorized as the open-loop system [10] and the closed-loop one [11]. Regarding the open-loop seeding rate control system, the drill velocity was measured by a ground wheel and a rotational encoder. On the basis of the drill velocity, the theoretical rotational speed of the electric motor can be calculated by the controller. Then, according to the theoretical rotational speed, the controller would control the electric motor to drive the seed-metering shaft with flute rollers. These flute rollers would discharge the seeds from the seed-metering device. On the basis of the open-loop system, the closed-loop seeding rate control system used an encoder to feedback the rotational speed of the flute roller and then control the rotational speed real-timely. The schematic diagrams of the open-loop and closed-loop seeding rate control systems are illustrated in Figure 1a,b.

At present, the open-loop seeding rate control system is widely used due to its simple physical implement. Theoretically, the rotational speeds of the flute roller would be adjusted according to the drill velocity and the theoretical seeding amount. Nevertheless, the control effect of the open-loop system cannot be assessed real-timely, because no variable was used to feedback to the controller system. Alameen et al. [12] developed the open-loop system which regulated the application rates of granular fertilizer by changing the position of the adjustment lever. The movement of the adjustment lever was controlled by a double-acting pneumatic cylinder. The theoretical fertilizer rate at each of lever positions was calibrated before experiments. Due to no feedback device of the position of the adjustment lever, the error of the position cannot be measured and compensated. The experimental results showed that the overall application rate error of the system was ±2.6%. Zhai et al. [13] used a stepper motor to replace the mechanical transmission system for regulating seeding rates for reducing the effect of the slipping of the ground wheel. This study reported that the open-loop systems could reduce the malfunctions caused by the mechanical transmission system. In practical sowing processes, if the contact surface of seed-metering shaft and machine is rusty or the error of the installation position of flute rollers was obvious, the actual rotational speed of flute rollers may not be equal to the theoretical value due to the large friction torque. In order to overcome this problem, the closed-loop control system for the roller speed was proposed. A Hall sensor or an encoder was used to monitor and feedback the rotational speed of the flute roller. According to the monitored values of the roller speed, the controller can regulate the speed of the flute roller real-timely. The schematic diagram of the conventional roller speed control system is displayed in Figure 1b. Besides, some controlling and optimizing algorithms, such as the Mamdani fuzzy logic reasoning and the genetic optimization, were applied to regulate the roller speeds [14,15]. Recently, by applying the aforementioned control algorithms, the closed-loop control systems have made an achievement in improving the accuracy of the roller speeds.

As a universal control algorithm, the proportional-integral-differential (PID) control algorithm had been applied to the roller speed control systems to reduce the errors between the actual roller speed and the theoretical one [16]. A study by Kamgar et al. [17] developed a roller speed control system for a conventional grain drill. This study applied the PID algorithm to control the roller speed. Their experimental results showed that the developed roller speed control system can reduce the coefficient of variation of the seeding rates from 3.443% to 1.724%. Moreover, Jafari et al. [18] designed a closed-loop roller speed control system based on the PID algorithm to regulate the roller speeds. They analyzed that the transition processes of the seeding rates rising from 87.5 kg ha^−1^ to 262.5 kg ha^−1^ and declining from 262.5 kg ha^−1^ to 87.5 kg ha^−1^. The experimental results showed that the response time of the above transition processes were 7.4 s and 5.2 s, respectively.

Based on the above discussion, it can be said that almost all research applied the rotational speed of flute roller as the indirect variable to control the actual seeding rates. The researchers in this field believed that the relationship of the roller speed and the seeding rate is linear, because in most cases the seeds’ mass discharged per rotation of the flute roller (SMDPRFR) can be regarded as a constant value. Although the accuracy of the roller speed is improved using the closed-loop system, but there are still some problems in practical sowing processes. As the seeding proceeds, the mass of the seeds in the hopper would decrease, resulting in the SMDPRFR declining. Furthermore, if the working length of the flute roller changes in the seeding process, the SMDPRFR would also be changed. Under these conditions, even if the roller speed is accurate, the errors of actual seeding rates still exist. However, if the seeding rate can be fed back directly, the mentioned drawbacks will be overcome. Thus, we proposed an adaptive roller speed control method (ARSCM) based on the monitoring values of real-time seed flow rates as an inner closed-loop to reject nonlinearities and disturbances such as hopper seed fullness etc. In the previous study, we developed a seed flow rate sensor which can be used under the large seeding rate condition [19]. This sensor was based on the optical seed detectors counting all seeds travelling down the duct from the flute roller to the soil chisel. Besides, the detection performance of the developed sensor had been validated, but it was not integrated into any roller speed control system before. In this study, it was the first time to use the self-developed seed flow rate sensor to directly monitor the seeding rate and control the seed rate based on its monitoring values. The seeding performance of the conventional roller speed control method (CRSCM) was compared with that of the ARSCM.

Furthermore, the objective of this study was to develop a close-loop seeding rate control method which can eliminate the disturbances caused by the reduction of the seeds’ mass in the hopper and the change of the working length of the flute roller. In order to achieve above objective, the hardware of the adaptive roller speed control system with the self-developed seed flow rate sensor was designed and the ARSCM was built to regulate the seeding rate in each sowing plot. Additionally, the comparison experiments were carried out between the ARSCM and the CRSCM. The experimental results were compared under the constant-velocity condition and the variable-velocity condition.

The rest of the paper is organized as follows: Section 2 describes the ARSCM based on the monitoring value of the actual seeding rate. Moreover, the experimental design and processes of the ARSCM and the CRSCM are also demonstrated in the Section 2. In the Section 3, the seeding performance of the ARSCM and that of the CRSCM are compared and analyzed. Moreover, Section 4 presents the conclusions of the research.

## 2. Materials and Methods

### 2.1. Hardware

The hardware system of the ARSCM was composed of a master computer (A55V, ASUSTeK Computer Inc., Taipei, China), an embedded controller (Elite, Xingyi electronic technology Co., Ltd., Guangzhou, China), a drill velocity sensor (E40S8, AUTONICS, Pusan, Republic of Korea) and several submodules for seed-metering unit controlling. For every seed-metering unit, it had a seed-meter in unit control submodule which included a stepper motor (57EBP98ALC, Times Brilliant LLC., Beijing, China), a stepper motor driver (HBS57, Times Brilliant LLC., Beijing, China), and a self-developed seed flow rate sensor. By this means, the flute roller in each seed-metering device can be controlled individually. The schematic view of the adaptive roller speed control system for a seed-metering unit is shown in Figure 2.

In detail, a human machine interface (XCOM V2.0, Guangzhou Xingyi electronic Co., Ltd., Guangzhou, China) was installed on the master computer to display the operational parameters of the developed system. The embedded controller whose core component was a micro control unit (MCU) (STM32F103ZET6, STMicroelectronics Co. Ltd., Geneva, Switzerland) was used to detect and control the real-time seeding rates. The embedded controller communicated with the master computer via the RS232 communicating protocol. Moreover, an incremental encoder was connected with the shaft of the ground wheel to monitor the drill velocity and the real-time seeding rate was determined according to the monitored drill velocity. Additionally, a speed sensor was mounted at the end of the motor to feedback the motor speed to the stepper motor driver.

A self-developed seed flow rate sensor, used to monitor the actual seeding rate, was installed in each seed-metering unit. The seed flow rate sensor was based on a seed flow reconstruction technique [19]. When a dense seed flow impacted on the dispersing boards, the seed flow could be uniformly separated as individual particles. Afterwards, these particles could fall into different channels and reconstruct several intermittent seed flows. There was an infrared ray in each channel. The number of seeds can be judged according to the time when the falling seeds shaded the infrared ray. After a specific time period, the number of seeds in each channel would added together as the actual seeding rate.

### 2.2. Adaptive Roller Speed Control Method (ARSCM)

The schematic view of the ARSCM is illustrated in Figure 3. Unlike the conventional seeding rate control strategies [20,21], the whole sown field in this study can be virtually segmented as small sowing plots whose length and width were a specific feedback distance and the width of the grain drill. The specific feedback distance would be determined by an optimal experiment whose results were described in the Section 3.1 and it did not vary with the speed of the tractor. The ARSCM can regulate the seeding rate in the (*i*+1)th sowing plot according to the actual seeding rate in the *i*th sowing plot (*i* is the serial number of the sowing plots). Controlling the seeding rates of every seed-metering unit in each sowing plot can effectively improve the seeding accuracy and seeding uniformity in the holistic sown area.

The block diagram of the ARSCM is displayed in Figure 4. First, the structural and operational parameters of the grain drill should be inputted in the theoretical seeding rate decision model. These parameters included the theoretical seeding amount per hectare, the working width of the grain drill, the length of a feedback distance (FD), the number of seeding units and the theoretical SMDPRFR. The theoretical seeding rate of a seeding unit in every sowing plot was calculated by the Equation (1):(1)$$S_{t}=\frac{Q\cdot W\cdot L}{10\cdot N}$$
where *S_t_* is the theoretical seeding rate of a seed-metering unit in a sowing plot, g; *Q* represents the theoretical seeding amount per hectare, kg ha^−1^; *W* denotes the width of the grain drill, m; *L* indicates the length of the feedback distance, m; *N* is the number of the seeding units of a grain drill. The *S_t_* was a constant value once it was calculated.

When the grain drill traveled the *i*th sowing plot, the seed flow rate sensor can detect and transmit the actual seeding rate to the embedded controller. Afterwards, the error of the seeding rate in the *i*th sowing plot can be calculated by the following Equation (2):(2)$$E_{sj}^{i}=S_{t}-S_{aj}^{i}$$
where $S_{aj}^{i}$ denotes the actual seeding rate discharged by *j*th seed-metering unit in the *i*th sowing plot, g; $E_{sj}^{i}$ is the absolute error of the seeding rate of the *j*th seed-metering unit in the *i*th sowing plot, g. In this study, the value range of the *j* was from 1 to *N*.

According to the magnitude of the $E_{sj}^{i}$, an expert control algorithm can calculate the regulating seeding rate, which can compensate the $E_{sj}^{i}$. After that, the grain drill could implement the sowing process according to the regulating seeding rate.The detailed operational processes how the regulating seeding rate can be obtained using the expert control algorithm would be explained in the Section 2.3.

Furthermore, a seeding rate to roller speed converting model can transfer the regulating seeding rate into the theoretical roller speed, which is demonstrated as follows Equation (3):(3)$$N_{tj}^{i+1}=\frac{S_{rj}^{i+1}\cdot V^{i}}{L\cdot q_{t}},$$
where $N_{tj}^{i+1}$ indicates the theoretical roller speed of the *j*th seed-metering unit in the (*i*+1)th seeding plot, g; $S_{rj}^{i+1}$ is the regulating seeding rate of the *j*th seed-metering unit in the (*i*+1)th sowing plot, g; *V^i^* represents the average drill velocity in the *i*th sowing plot, m s^−1^; *q_t_* denotes the theoretical SMDPRFR, g rev^−1^.

The motor speed was determined by the transmission ratio between the flute roller and stepper motor. A proportional-integral algorithm, which already was programmed in the stepper motor driver, was used to control the motor speed according to the error of roller speed (*E_nj_*). Once the flute roller was driven, grain seeds could be pushed out of the seed-metering devices. The actual seeding rate was the product of the actual SMDPRFR, the actual roller speed and the traveling time. From another perspective, the actual seeding rate can be viewed as the theoretical seeding rate plus a seeding-rate error, which can be illustrated by Equation (4):(4)$$S_{aj}^{i+1}=q_{aj}^{i+1}\cdot N_{aj}^{i+1}\cdot \frac{L}{V^{i+1}}=S_{t}+E_{sj}^{i+1}=q_{t}\cdot N_{tj}^{i+1}\cdot \frac{L}{V^{i+1}}+E_{sj}^{i+1}$$
where $S_{aj}^{i+1}$ is the actual seeding rate of the *j*th seed-metering unit in the (*i*+1)th sowing plot, g; $q_{aj}^{i+1}$ represents the actual SMDPRFR of the *j*th seed-metering unit in the (*i*+1)th sowing plot, g rev^−1^; $N_{aj}^{i+1}$ refers to the actual roller speed of the *j*th seed-metering unit in the (*i*+1)th sowing plot, rev s^−1^; *V^i+1^* represents the average drill velocity in the (*i*+1)th sowing plot, m s^−1^; $E_{sj}^{i+1}$ denotes the seeding-rate error of the *j*th seed-metering unit in the (*i*+1)th sowing plot, g.

After the grain drill travelled a sowing plot, the above ARSCM would be executed by *N* times to regulate the seeding rate of every seed-metering unit.

### 2.3. Expert Control Algorithm

Caused by the structure of the flute roller, the actual seeding rate may have a random error whose magnitude and sign were uncertain [15]. However, the random error was small but difficult to be eliminated. Therefore, in order to avoid frequent adjustments, a slight fluctuation of actual seeding rates was acceptable. Once the seeding rates was subject to steady disturbances, the error of the actual seeding rate could exceed the limit of the random error and thus seriously affect the seeding accuracy. At this time, the seeding rate should be regulated.

Taking the above control requirements into consideration, the ARSCM should have an ability to intelligently determine whether the seeding rate needed to be adjusted or not. In other fields, the expert control algorithm was widely used in uncertain, inaccurate and nonlinear control processes [22,23,24,25]. According to the characteristics of the sowing, the expert control algorithm was appropriate to be applied in this research. The schematic view of the expert control algorithm is illustrated in Figure 5.

As seen in Figure 5, the expert algorithm was consisted of five modules, namely, a relative error calculation module, an inference engine, a knowledge base, a compensation increment calculation module and a regulating seeding rate calculation module.

Moreover, the aim of the expert control algorithm was to obtain the regulating seeding rate of the *j*th seed-metering unit in the (*i*+1)th sowing plot. The regulating seeding rate can compensate the seeding-rate error resulted from the steady disturbance. In this study, according to the relative error of the actual seeding rate, the compensation coefficient was determined. However, the relative error was computed by the following Equation (5).
(5)$$RE_{sj}^{i}=\frac{\left|E_{sj}^{i}\right|}{S_{t}}\times 100\%,$$
where $RE_{sj}^{i}$ is the relative error of the seeding rate discharged by *j*th seed-metering unit in the *i*th sowing plot, %.

Afterwards, the $RE_{sj}^{i}$ would be sent to the inference engine, which included four inference rules. The compensation coefficient can be inferred according to the magnitude of the $RE_{sj}^{i}$. These inference rules are listed as follows:**Rule 1: If** (0 < $RE_{sj}^{i}$ ≤ EB_1_), **Then**
*P* = 0;**Rule 2: If** (EB_2_ ≥ $RE_{sj}^{i}$) **and** ($RE_{sj}^{i}$ > EB_1_), **Then**
*P* = M_1_;**Rule 3: If** (EB_3_ ≥ $RE_{sj}^{i}$) **and** ($RE_{sj}^{i}$ > EB_2_), **Then**
*P* = M_2_;**Rule 4: If** (EB_3_ < $RE_{sj}^{i}$), **Then**
*P* = M_3_;
where the EB_n_ represents the *n*th error band; *P* indicates the compensation coefficient. The magnitude order of the error bands was 0 < EB_1_ < EB_2_ < EB_3_. Regarding the compensation coefficient, it can be assigned as the one of 0, M_1_, M_2_ or M_3_.

In order to ascertain the specific values of the error bands and the compensation coefficients, preliminary tests were carried out. In these tests, the CRSCM, as same as the method written in the Section 2.4.1., was used to regulate seeding rates. According to the results of preliminary tests, we found that in the common testing conditions (no steady disturbance), the most relative errors of the seeding rates were less than 5%. When the mass of seeds in the hopper was insufficient, the steady error was between 5% and 15%. Thus, the EB_1_, EB_2_ and EB_3_ were set as 5%, 10% and 15%, respectively. Regarding the compensation coefficients, they can determine the regulative intensity of the seeding rate. The initial values of the compensation coefficients were from 0.05 to 1 with an interval of 0.05. According to the results of cut-and-try tests, the M1, M2 and M3 were set as 0.1, 0.2 and 0.5, respectively.

After ascertaining the compensation coefficient, the compensation increment can be computed by inputting the compensation coefficient into the compensation increment calculation module (i.e., Equation (6)).
(6)$$IN_{j}^{i+1}=P\times E_{sj}^{i},$$
where *IN_j_^i+1^* represents the compensation increment for the seeding rate of the *j*th seed-metering unit in the (*i*+1)th sowing plot, g.

The regulating seeding rate in the (*i*+1)th sowing plot was obtained via adding the compensation increment and the *i*th regulating seeding rate together. This procedure can be done using the regulating seeding rate calculating module, as seen in Equation (7).
(7)$$S_{rj}^{i+1}=S_{rj}^{i}+IN_{j}^{i+1},$$
where $S_{sj}^{i}$ and $S_{sj}^{i+1}$ are the regulating seeding rates of the *j*th seed-metering unit in the *i*th and (*i*+1)th sowing plot, g.

### 2.4. Performance Comparison between the ARSCM and the CRSCM

#### 2.4.1. Conventional Roller Speed Control Method (CRSCM)

In practical sowing operation, the CRSCM was widely used to control the seeding rates. The object of the CRSCM was to regulate the rotational speed of the flute roller approaching to the theoretical value. The hardware system of the CRSCM was consisted of an embedded controller, a stepper motor driver and a stepper motor. The CRSCM used a seed-metering shaft to drive all flute rollers, so that all of these flute rollers had a common rotational speed. The theoretical value of the roller speed was calculated by the theoretical roller speed decision model (i.e., Equation (8)).
(8)$$N_{t}=\frac{Q\cdot W\cdot V^{i}}{10\cdot N\cdot q_{t}},$$
where *N_t_* is the theoretical rotational speed of the flute roller, rev s^−1^.

Due to the frictional torque between the flute roller and the seed-metering device, the actual roller speed was not equal to the theoretical one. In this study, the roller speed and the stepper motor speed were equal, because they were connected directly by a shaft coupling. A proportional-integral algorithm was used to control the rotational speed of the motor. The regulated motor speed can be obtained by the following Equation (9).
(9)$$N_{r}^{k+1}=K_{P}\cdot \left[E\left(k\right)-E\left(k-1\right)\right]+K_{I}\cdot E\left(k\right)+N_{r}^{k},$$
where *K_P_* and *K_I_* are the proportional coefficient and the integral coefficient, respectively; *E(k)* denotes the error of the motor speed at the *k*th sampling points. *N_r_^k+1^* and *N_r_^k^* represent the regulating rotational speed of the flute roller at (*k*+1)th and *k*th sampling point, rev s^−1^.

The parameters of the proportional-integral algorithm were the defaults, because they were set as the fixed values by the manufacturer of the stepper motor. The block diagram of the CRSCM is illustrated in Figure 6. Note that the actual seeding rates were monitored and displayed on the human machine interface but were not used as the feedback values.

#### 2.4.2. Experimental Equipment and Evaluation Index

In order to evaluate the seeding performance of the ARSCM and that of the CRSCM, comparison experiments were carried out on a fabricated seed-metering test platform [26], as seen in Figure 7. In this way, the experimental levels can be changed efficiently, and the non-experimental variables can be effectively controlled. The seeding data were uploaded to the master computer from the embedded controller. These data included the serial number of sowing plots, the theoretical seeding rate, the actual seeding rate, the regulating seeding rate and the uploading time. The sampling period was equal to the time when the virtual grain drill traveled a sowing plot. The serial communication software XCOM was used to display the uploaded seeding data and save them in TXT files.

In this research, wheat seeds were used, and we measured 100 wheat seeds to obtain their average dimensional sizes. Their average length, width and height were 6.17 mm, 3.28 mm, and 2.96 mm, respectively, and the thousand seed mass of the seeds was 46.0 g. Moreover, the length, width and height of the seeds’ volume in the hopper was 250 mm, 50 mm and 45 mm, respectively. The experiment would stop until no seeds were discharged out of the seed-metering device.

Because the volume of seeds in the hopper was a certain value no matter under what experimental condition, the higher the seeding amount per hectare, the shorter the total traveling distance of the virtual grain drill. In addition, under the same seeding amount per hectare and the same drill velocity, the total traveling distance in each experimental repetition could be slightly different, because the experimental sowing processes could be same in the different experimental repetitions.

Regarding other drill parameters used in the experiments, the working width, the SMDPRFR and the number of seeding units of a grain drill were set as 2.4 m, 14 g rev^−1^ and 14, respectively. These parameters were as same as those of an off-the-shelf grain drill (2BFG-12, Jiangsu Xintian mechanical manufacturing Co., Ltd., Zhenjiang, China).

In order to assess the seeding accuracy and seeding uniformity of the two methods, two indicators were used. An indicator was the accuracy of seeding rates (*SA*), and the other was the coefficient of variance of seeding rates (*SCV*). The computational methods of the above indicators were illustrated in Equations (10) and (12):(10)$$SA=\frac{1}{n}\cdot \left[\sum_{i=1}^{n}\left(1-\left|\frac{S_{a}^{i}}{S_{t}}-1\right|\right)\times 100\%\right],$$
(11)$$SD=\sqrt{\frac{1}{n}\cdot \sum_{i=1}^{n}{\left(S_{a}^{i}-S_{aver}\right)}^{2}},$$
(12)$$SCV=\frac{SD}{S_{aver}}\times 100\%,$$
where *SD* represents the standard deviation of all seeding rates in a repetition, g; *S_aver_* is the average value of all seeding rates in a repetition, g; *n* represents the max serial number of sowing plots.

#### 2.4.3. Comparison Experiment under Constant-Velocity Conditions

In order to evaluate the seeding performance of the ARSCM and that of the CRSCM, comparison experiments were carried out under the constant-velocity conditions. The virtual grain drill would maintain the same velocity in each experimental repetition.

Regarding the ARSCM, the seeding rate was determined, based on the seeding amount per hectare, the drill velocity and the length of the feedback distance. In addition, in Jiangsu province, the value interval of the seeding amount is 150–225 kg ha^−1^ and that of the drill velocity is 0.83–1.38 m s^−1^. Concerning the length of the feedback distance, Tola et al. [20] acclaimed that 4 or 5 m is the optimal length when the range of the tested length will be between 1–5 m. Nevertheless, they did not study the performance of their developed system with longer feedback distances. In order to study the seeding performance of the ARSCM with a wider range of the feedback distance, the levels of the feedback distance were set as 2.5 m, 5 m and 7.5 m.

Also, the constant-velocity experiments were designed as the full-factorial experiments. Taking experimental quantity into consideration, each factor had three levels and every experiment was replicated three times. The levels of each factor are listed in Table 1. In the following section of this paper, the seeding amount per hectare, the drill velocity and the feedback distance could be represented by the Q, V and FD, respectively. Moreover, a level of an experimental factor could be defined as Qnum, Vnum and FDnum, where num refers the specific value of an experimental level. By this way, a level combination of the experiments can be expressed as the following form: Qnum-Vnum-FDnum.

Furthermore, the procedure of the constant-velocity experiments was expounded as follows. Firstly, the full-factorial constant-velocity experiments of the ARSCM was conducted. According to the results of the above experiments, the optimal feedback distance would be ascertained. As a comparison, the seeding rate controlled by the CRSCM was also monitored in each optimal feedback distance. The CRSCM carried out under the same level-combinations with the ARSCM, whose experimental factors were the seeding amount per hectare and the drill velocity.

#### 2.4.4. Comparison Experiment under Variable-Velocity Conditions

In practical seeding processes, the drill velocity was unsteady due to slippages. In order to simulate the more practical seeding processes, the laboratory experiments were conducted under the variable-velocity conditions. That is to say, the flute roller would frequently change its theoretical rotational speed according to the velocity of virtual grain drill. The theoretical roller speed of the ARSCM and that of the CRSCM can be calculated by using Equations (3) and (8), respectively. Moreover, the experimental factor of the variable-velocity conditions was the seeding amount (per hectare) whose levels were 150 kg ha^−1^, 187.5 kg ha^−1^ and 225 kg ha^−1^. In these experiments, the feedback distance was applied as the optimized one. Both of the ARSCM and the CRSCM were tested under the above experimental level conditions. Each experiment was performed three times.

The variable-velocity experimental process can be expounded as follows. At first, the virtual grain drill was assumed to travel in ten feedback distances at the middle velocity (1.11 m s^−1^). Then, the virtual grain drill was assumed moving at variable velocity processes, including a high velocity process (1.38 m s^−1^), a middle velocity process (1.11 m s^−1^) and a low velocity process (0.83 m s^−1^), and each process lasted five sowing plots. Afterwards, the virtual grain drill would sow at the middle velocity (1.11 m s^−1^) until no seed was discharged out of the seed-metering device. The schematic view of the variable-velocity experimental processes is shown in Figure 8.

### 2.5. Statistical Analysis

A commercial statistical software IBM SPSS Statistics (21.0.0.0, International Business Machine Corp., Armonk, NY, USA) was applied to perform the analysis of variance (ANOVA). The ANOVA was used for evaluating whether a factor can significantly influence the SA and the SCV at 0.005 significant level.

## 3. Results

### 3.1. Determination of the Optimal Feedback Distance

The SA values and SCV values of the ARSCM with different feedback distances are shown in Figure 9a,b, respectively. The range of the SA was from 84.57% to 94.59%, while the SCV ranged from 5.23% to 19.00%.

As seen in Figure 9, the ARSCM with the FD7.5 had the highest average SA (94.12%) and the lowest average SCV (6.77%). In most cases, the SA was raised with an increasing feedback distance, but the SCV had an inverse trend. However, with the ARSCM at the feedback distance of 2.5 m, the average SA and SCV were 87.94% and 15.28%, respectively, and at the feedback distance of 5 m, the average SA and SCV were 93.95% and 7.35%, respectively. As seen in Figure 9a, under the same seeding amount and the same velocity, the SA values of the FD5 and the FD7.5 did not have a significant difference, but both values were significantly different with the SA value of the FD2.5. Thus, the same conclusion can be drawn from the results of the SCV values, which can be seen in Figure 9b.

Under the same seeding amount, it was noted that when the feedback distance was short, the number of the roller rotations were relatively small. The error of the seeds’ mass in a roller rotation, resulted from the structure of the flute roller, may not be counteracted by another one. Therefore, the error of the seeding rate in each sowing plot was relatively large. The seeding performance of the ARSCM with the FD2.5 was the worst and we do not suggest to select the feedback distance less than 5 m.

Moreover, in most cases, the observed optimal feedback distance was 7.5 m, which can be viewed in Figure 9. This can be explained by the two following reasons.

When the seeding rate was regulated by the expert control algorithm, a transition process of the seeding rate can occur at the initial section of a new sowing plot [27]. Compared to 2.5 m and 5 m, if 7.5 m was applied as the feedback distance, the ratio of the transition distance to the feedback distance was the least. In this way, the influence of the transition process of the seeding rates on the seeding performance can be reduced.Moreover, if the feedback distance was set as 7.5 m, the flute roller can revolve more rotations when the drill traveled a sowing plot. The random errors of the actual SMDPRFR had more chances to be offset because some random errors had plus signs while the others had negative signs. Therefore, the sum of these random errors in a sowing plot (i.e., the error of the seeding rate) was relatively small. Accordingly, the seeding performance of the ARSCM with the FD7.5 was the best.

When the seeding amount was 225 kg ha^−1^ and the feedback distance was 7.5 m, the theoretical seeding rate in each sowing plot was the largest compared with the other experimental conditions. Although the ARSCM can provide a compensation increment to the seeding rate, the decline speed of seed-mass in the hopper was still too fast, thereby affecting the accuracy of the seeding rate. In the initial section of a sowing plot, the seeding rate was acceptable due to the compensation, but in the rest part, the seeding accuracy was subject to the declining seed-mass. Essentially, when the seeding amount was 225 kg ha^−1^ and the feedback distance was 7.5 m, the regulative frequency was so low that the ARSCM was not sensitive to the steady errors of seeding rates. Under the condition of Q225-V1.38, the SA of the feedback distance of 5 m was even higher than that of the feedback distance of 7.5 m. Hence, the feedback distances more than 7.5 m were not suggested in the high seeding amount.

### 3.2. Performance Comparison between the ARSCM and the CRSCM under the Constant-Velocity Conditions

When the feedback distance was set as 7.5 m, the SA values of the ARSCM and the CRSCM are shown in Figure 10a and the SCV values of these two methods are illustrated in Figure 10b. The SA values of the ARSCM ranged in the interval from 93.08% to 94.59% and those of the CRSCM was from 88.34% to 90.31%. The SCV interval of the ARSCM was 5.23–7.85% and that of the CRSCM were 7.76–10.67%. Under the same seeding amount and the same drill velocity, the SA value of the ARSCM was significantly higher than that of the CRSCM. However, the SCV values of these two methods were marked with the same letters except the experiments of Q225-V1.11 and Q187.5-V1.38. This indicated that in most cases, the seeding uniformities of the two methods was not statistically different at 0.05 level.

In order to analyze the seed distribution along the whole virtual traveling distance, the results of seeding rate trends of the ARSCM and the CRSCM are shown in Figure 11. In each subplot, every point was the average seeding rate of the three replicates. When the seeding amount were 150 kg ha^−1^, 187.5 kg ha^−1^ and 225 kg ha^−1^, the theoretical seeding rates were 19.29 g, 24.11 g and 28.93 g, respectively.

As for the ARSCM, the seeding rates gradually increased in the first 50 m. Afterwards, the seeding rates were in the steady state and they hovered around the theoretical value until no seed was discharged out of the seed-metering device. During the sowing process, once the seeds’ mass in the hopper was insufficient, the steady error of seeding rates would appear. The ARSCM can compensate the steady errors of seeding rates by increasing the regulating seeding rates. Thus, the seeding rates could not decrease as the increasing traveling distance. As for the seeding rates controlled by the CRSCM, the roller speed was accurate, but the SMDPRFR could decrease as the mass of the seeds in the hopper dwindled. When the traveling distance was from 0 to 100 m, the seeding rates were steady. However, the seeding rates would gradually decline as the sowing process implementing. This can explain why the seeding performance of the ARSCM was better than that of the CRSCM.

### 3.3. Performance Comparison of the ARSCM and the CRSCM under Variable-Velocity Conditions

Figure 12 displays that the SA and SCV values of the two control methods which were tested under the variable-velocity conditions. As seen in Figure 12, under the same seeding amount, the SA value of the ARSCM was higher than that of the CRSCM, but the SCV values of these two methods had an inverse tendency. The average SA and SCV of the ARSCM were 91.58% and 11.08%, respectively, while those of the CRSCM were 88.48% and 13.08%, respectively.

Figure 12a reveals that under the same seeding amount, the SA of the ARSCM was significantly better than that of the CRSCM at 0.05 level. However, as seen in Figure 12b, the SCV of the ARSCM had at least one same letter with that of the CRSCM. This meant that the seeding uniformity of the ARSCM and that of the CRSCM were not significantly different on statistics.

In addition, Figure 13 displays the seeding rate trends of the ARSCM and that of the CRSCM under the variable-velocity conditions. Each marker indicates the average value of three replications.

There were three rules which can be inferred from Figure 13.

From 0 to 50 m, the seeding rates controlled by the ARSCM were less than those controlled by the CRSCM. Theoretically, in the fourth sowing plot, the regulating seeding rate would reach the 90% of the theoretical seeding rate. In contrast, the seeding rate controlled by the CRSCM could be close to the theoretical value directly. Thus, in the initial four sowing plots, the seeding rates of the ARSCM were lower than those of the CRSCM.No matter which control method was applied, the seeding rate could vary as the drill velocity changed. The curves in Figure 13 reveals that both of these two control methods were significantly affected by the varying drill velocity.After 200 m, the seeding rates controlled by the CRSCM decrease sharply, while the seeding rates controlled by the ARSCM was still steady. This can prove that under the variable-velocity conditions, the ARSCM can effectively control the seeding rates, while the CRSCM cannot achieve that.

## 4. Discussion

The conventional seeding machine CRSCM with a machine speed wheel feedback is not open loop but is closed loop control, with either mechanical or electrical feedback of machine speed, to adjust the flute roller speed following the machine ground speed. Of course, ground speed today can be measured also with other type of sensors but DGPS is not a favorable solution to achieve such accuracy that other soil conditions disturbances make it not worth to achieve. The seed measuring sensor introduced in this article is another inner closed loop control to reject the claimed disturbances of the hopper seed mass and of the flute roller, which is the final control element (FCE). Of course, in such precision there is a question if it is worth to also have individual flute roller control instead all together on one shaft. Because if one sensor represents all rollers then there may be differences on each line sown.

### 4.1. Advantages of the ARSCM

In this study, the ARSCM, which was based on the monitoring value of the seeding rates, was proposed. Unlike the conventional CRSCM, the ARSCM used the actual seeding rates as the feedback values along with the roller speeds. Compared to the CRSCM, the advantages of the ARSCM were listed as follows:Seed flow rate sensors were integrated into the control system and thus actual seeding rates can be detected and fed back. In practice, seeds’ mass in the hopper would decline after a long sowing process. Sometimes, the seeds may move to one side of the hopper because of machine vibrations. These situations may lead to the steady error of the SMDPRFR. When using the CRSCM, the roller speed was close to the theoretical one, but the steady errors of the SMDPRFR would still exist. This would lead to the errors of the seeding rates. Nevertheless, when applying the ARSCM, the actual seeding rates can be monitored, and the steady error of the seeding rates can be compensated.Using the ARSCM, the seed distribution was more uniform in the whole sowing area. The CRSCM sets the seeding amount per hectare as the control objective. Sometimes, although the accuracy of seeding amount in the total sowing area is acceptable, the seeding accuracy in each sowing plot may not be ideal. In Figure 10b and Figure 12b, the SCV values of the ARSCM were lower compared to those of the CRSCM. It indicates that the seeding uniformity of the ARSCM was better than that of the CRSCM.

When the tested feedback distance was in the range of 2.5 m to 7.5 m, the results of the experiments showed that the longer the feedback distance the higher the SA value. A similar conclusion was obtained by Reyes et al. [28]. They tested the developed fertilizing control system with feedback distance between 1 m to 5 m and acclaimed that when feedback distance was selected as 5 m, the fertilizing accuracy of their developed system was the highest (in the middle of 86% to 88%). However, they did not discuss the performance of the developed system when feedback distance was longer than 5 m. In addition, they only provided the overall error in the total traveling distance but did not assess the seeding uniformity in the holistic fertilizing area.

### 4.2. Implication

The ARSCM not only can be used in agriculture processes, such as seeding and fertilizing, but also might be success in other fields where small particles need to be accurately discharged. For example, in the food and medical industries, the ARSCM can replace the conventional methods for bagging processes. Moreover, in the chemical field, the ARSCM can be used to quantificationally put micro chemical particles into the reactor.

### 4.3. Future Research

The ARSCM can accurately distribute seeds in each sowing plot because it can detect and adjust actual seeding rates real-timely. However, the performance of the ARSCM could be better if the following suggestions are accepted in the future:In the first five sowing plots, the open-loop method can be used to replace the ARSCM. As seen in Figure 11 and Figure 13, the rising speed of the seeding rates controlled by the ARSCM is slow. If an open-loop control method were applied in these sowing plots, the seeding rates could be close to the theoretical one directly and the transition process of the seeding rates could be accelerated.When the grain drill has traveled the *i*th sowing plot, the errors of seeding rates in the previous feedback distances could be taken into consideration for determining the regulating seeding rate in the (*i*+1)th sowing plot. If the seeding rates in the previous sowing plots were considered, the trend of the seeding rates can be predicted and thus the seeding performance of the ARSCM might be improved.In order to study whether the machine vibrations and the actual drill velocity could affect the SA and SCV of the ARSCM, the field experiments should be conducted in the next step.It is obvious that the expert control algorithm described in 2.3 (Figure 5) is actually an expert adaptable PID controller using an error *E* filter that is the feedback distance to reject the random noise influence. The feedback distance approach is not the best choice as a segmented period filtering. The filter will be replaced with matched short time running average and the expert system will adjust the running period or the forgetting factor of the filter. As regards the computation of the compensation factor *P* it will be better computed with an adaptable (staged or nonlinear) PID regulator.

## 5. Conclusions

The main success of this research was developing the ARSCM which can feedback and control the seeding rate directly. Both ARSCM and the CRSCM were compared under the constant-velocity conditions and the variable-velocity conditions. The following three conclusions were obtained by analyzing the experimental results:Under the constant-velocity condition, the performance of the ARSCM was validated with different feedback distances. According to the results, the performance of the proposed system was the best when the feedback distance was 7.5 m. Applying this feedback distance, the average SA and SCV of the ARSCM were 94.12% and 6.77%, respectively. When the ARSCM used 2.5 m as the feedback distance, its average SA and SCV were 87.94% and 15.28%, respectively; regarding the proposed system with the feedback distance of 5 m, its average SA and SCV were 93.95% and 7.35%, respectively. According to the above results, 7.5 m was determined as the optimal feedback distance.As a comparison, the performance of the CRSCM was also validated. The seeding rate controlled by the CRSCM was monitored in every 7.5 m. The average SA and SCV of these seeding rates were 89.00% and 8.95%, respectively. It showed that under the constant-velocity conditions, both of the SA and the SCV of the ARSCM were better than those of the CRSCM.Moreover, under the variable-velocity conditions, the average SA and SCV of the ARSCM were 91.58% and 11.08%, respectively; those of the CRSCM were 88.48% and 13.08%, respectively. These results indicated that under both of the constant-velocity and the variable-velocity conditions, the performance of the ARSCM was better than that of the roller speed control method.

In the overall next stage, control developments will have the smart control approaches and convert the machine to a smart seeder machine. Advanced PID controller approach with an expert system tunning the three parameters of the PID algorithm and with feedforward variables for the starting case (in replacement of equivalent open loop suggested before) will provide much better performance. A smart short time low pass filter of seed flow rate meter also will be replacing the feedback distance which will be also adaptable to the case covering disturbances of machine mechanical parts and soil conditions.

## Figures and Tables

**Figure 1 sensors-21-00080-f001:**
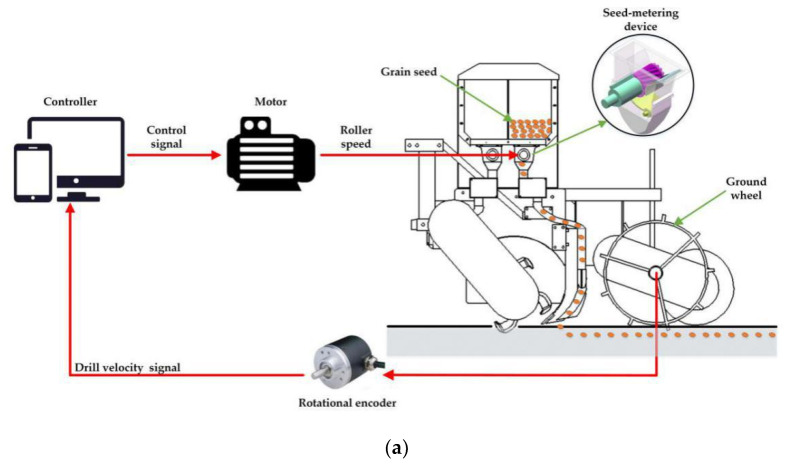
The schematic diagrams of the conventional roller speed control systems: (**a**) open-loop system; (**b**) closed-loop system.

**Figure 2 sensors-21-00080-f002:**
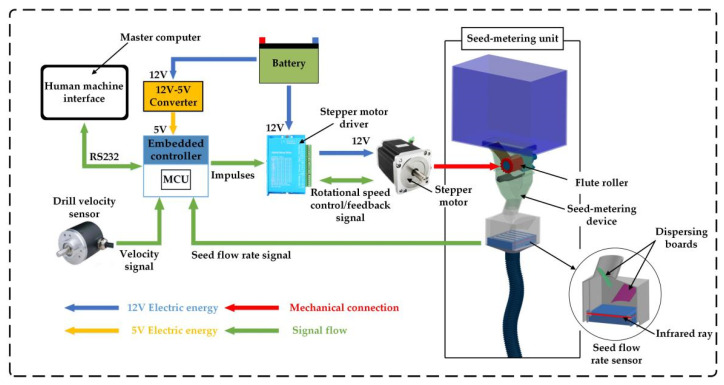
Schematic view of the adaptive roller speed control system for a seed-metering unit.

**Figure 3 sensors-21-00080-f003:**
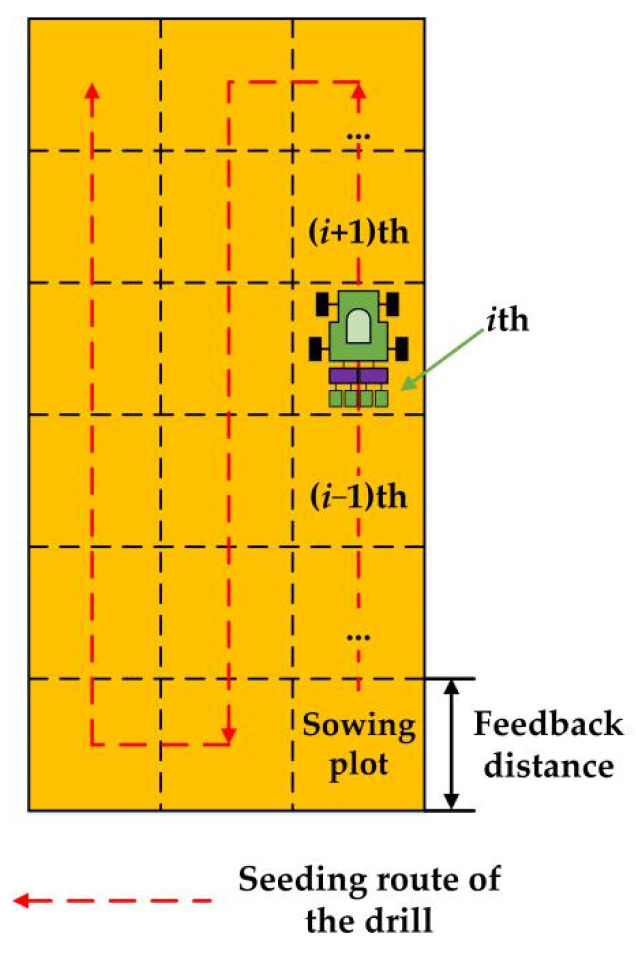
Schematic view of the ARSCM.

**Figure 4 sensors-21-00080-f004:**
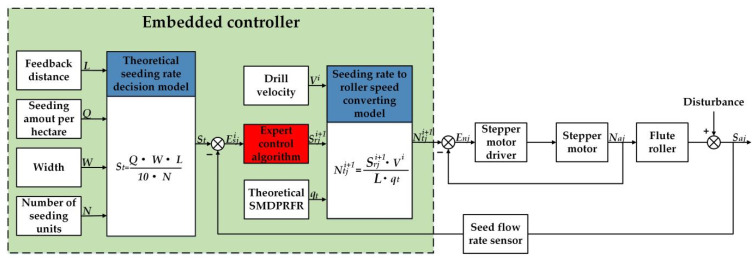
The block diagram of the ARSCM.

**Figure 5 sensors-21-00080-f005:**
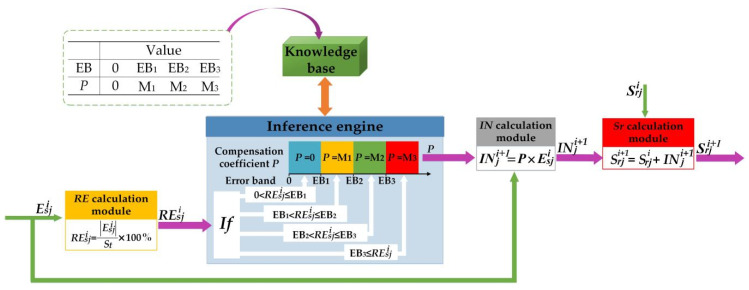
Schematic view of the expert control algorithm. *RE* means the relative error of the seeding rate; *IN* indicates the compensation increment of the seeding rate; and *Sr* represents the regulating seeding rate.

**Figure 6 sensors-21-00080-f006:**
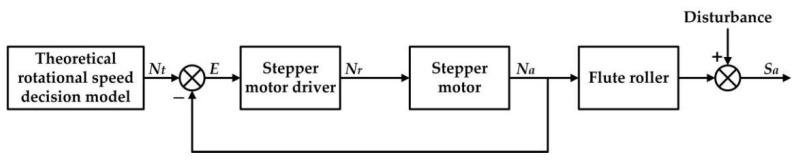
Block diagram of the CRSCM. *N_a_* is the actual roller speed, rev^−1^ s; *N_r_* refers the regulating roller speed, rev^−1^ s; *S_a_* indicates the actual seeding rate, g; *E* is the error of the roller speed, rev^−1^ s.

**Figure 7 sensors-21-00080-f007:**
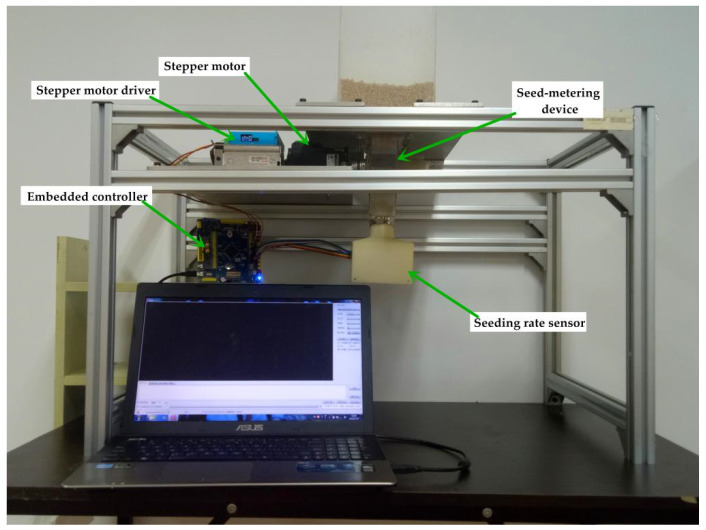
Fabricated seed-metering test platform.

**Figure 8 sensors-21-00080-f008:**
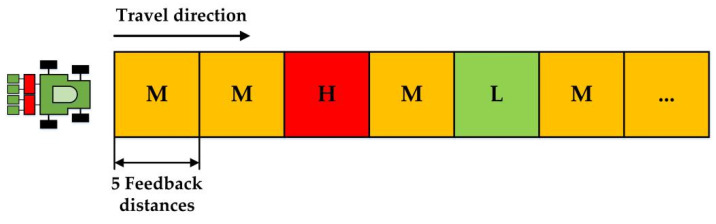
Schematic view of the variable-velocity experiment. Every square represents five sowing plots. The letter H, M and L denote the virtual distances where a drill travels at a high, middle and low velocity, respectively.

**Figure 9 sensors-21-00080-f009:**
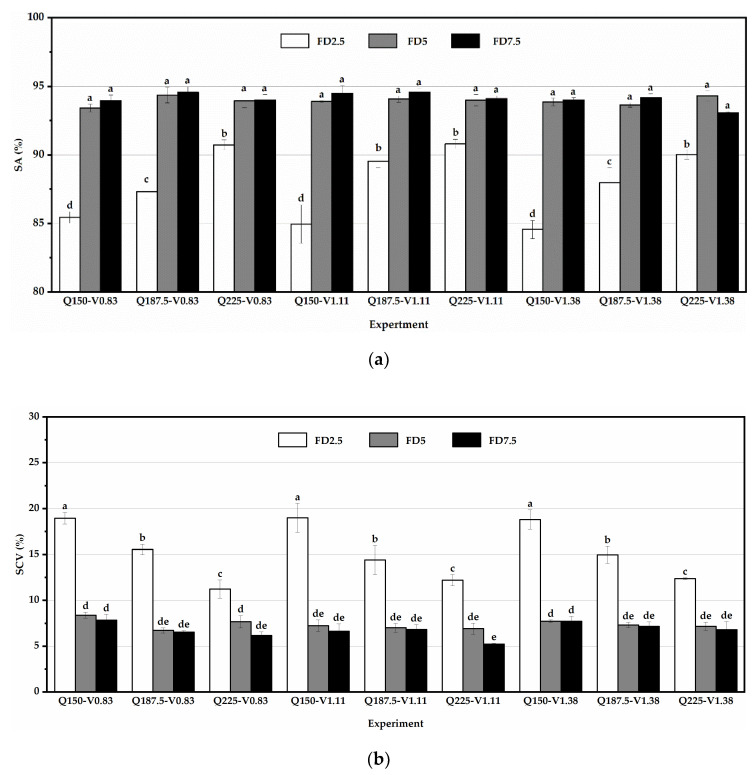
Performances of the ARSCM under different experimental conditions. (**a**) The SA values of the ARSCM with different feedback distances; (**b**) The SCV values of the ARSCM with different feedback distances. Q represents the seeding amount and V means the velocity; FD2.5, FD5 and FD7.5 represent the feedback distance of 2.5 m, 5 m and 7.5 m, respectively. Data represent by means ± standard errors. The letters (a, b, c, d and e) represent the significance between the results obtained in different experiments. If the letter on any two bars are different, it means they are significantly different.

**Figure 10 sensors-21-00080-f010:**
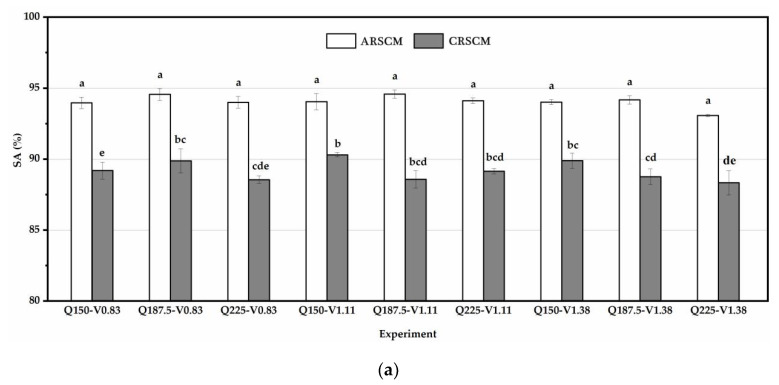
Seeding performances of the ARSCM and the CRSCM under constant-velocity conditions. (**a**) The SA values of the two methods. (**b**) The SCV values of the two methods. Q is the seeding amount per hectare; V means the drill velocity.The letters (a, b, c, d, e and f) represent the significance between the results obtained in different experiments. If the letter on any two bars are different, it means they are significantly different.

**Figure 11 sensors-21-00080-f011:**
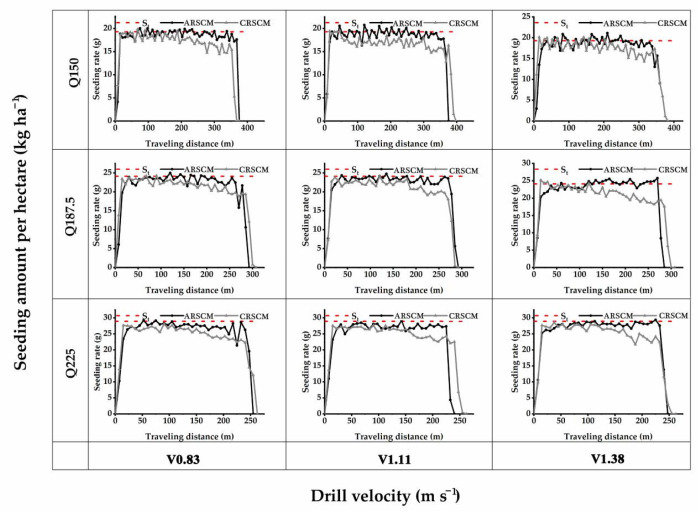
Trends of the seeding rates controlled by the ARSCM and the CRSCM. Q means the theoretical seeding amount per hectare, kg ha^−1^; V represents the drill velocity, m s^−1^; the number close to the Q or V denotes the specific value of experimental level of the Q or V; *S_t_* represents the theoretical seeding rate.

**Figure 12 sensors-21-00080-f012:**
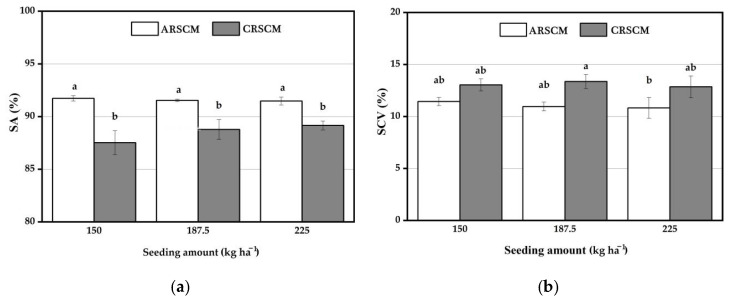
Seeding performances of the ARSCM and the CRSCM under the variable-velocity conditions. (**a**) The SA values of two methods. (**b**) The SCV values of two methods.The letters (a and b) represent the significance between the results obtained in different experiments. If the letter on any two bars are different, it means they are significantly different.

**Figure 13 sensors-21-00080-f013:**
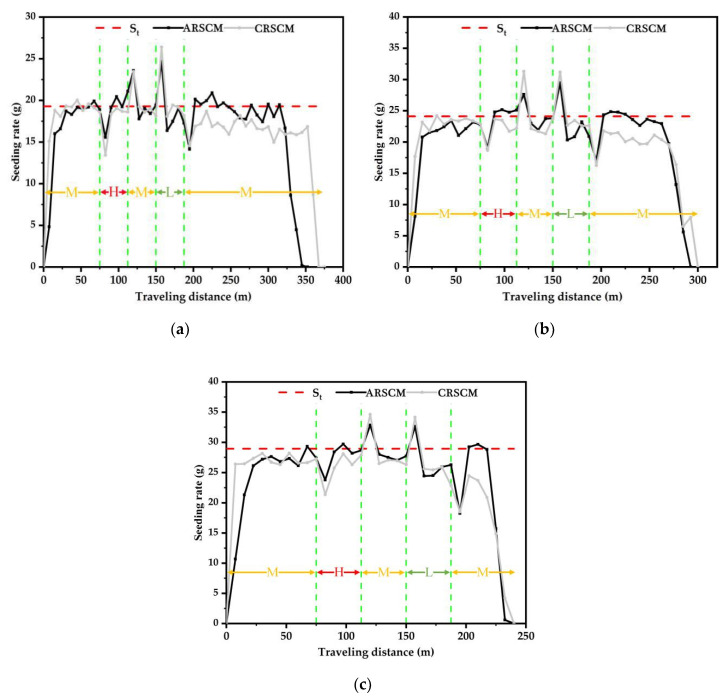
Seeding rate trends of the ARSCM and these of the CRSCM under the variable-velocity conditions. (**a**) The seeding rate trends of the two methods under the seeding amount of 150 kg ha^−1^. (**b**) The seeding rate trends of the two methods under the seeding amount of 187.5 kg ha^−1^. (**c**) The seeding rate trends of the two methods under the seeding amount of 225 kg ha^−1^. *S_t_* is the theoretical seeding rate. The letter L, M, and H indicate that the traveling distance where the virtual drill travels at the low, mediate and high velocity, respectively.

**Table 1 sensors-21-00080-t001:** The levels of each factor in the constant-velocity experiments.

Factor	Level		
	1	2	3
Seeding amount per hectare (Q, kg ha^−1^)	150	187.5	225
Drill velocity (V, m s^−1^)	0.83	1.11	1.38
Feedback distance (FD, m)	2.5	5	7.5

## Data Availability

The data presented in this study are available in [supplementary file whose name is ROW DATA].

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
