# Peer review of "An Adaptive Roller Speed Control Method Based on Monitoring Value of Real-Time Seed Flow Rate for Flute-Roller Type Seed-Metering Device"

_sensors, 2020, doi:10.3390/s21010080_

Round 1

Reviewer 1 Report

1、How is the feedback distance in Figure 2 determined? What is the relationship between the feedback distance and the size of the sowing plot and the speed of the tractor?

2、What is the working principle of the self-developed seeding rate sensor? The focus of this article should be on the sensor, otherwise the other results are meaningless.

3、Figure 6 is only a simulated test platform, which may be very different from the actual field operation. Field test data should be added in this article.

4、How is the RSC data obtained in the results? If the data of the two methods are not obtained under the same conditions, the comparison of the result analysis part will lose its significance.

Reviewer 2 Report

The article is devoted to the actual problem of real-time adjustment of the seeding rate. Undoubtedly, this problem occurs wherever there is a fairly large volume of sown seeds. The costs of overspending seeds are quite noticeable, moreover, sowing two or more seeds in one place makes it difficult to grow and causes competition for plants.

In my opinion, the article needs some improvement.

First, title
"A Seeding Rate Control Approach Based on the Monitoring Values of Real-time Seeding Rates"
assumes that there is some approach (which can be used on all seeding machine designs without exception), but in the introduction you can only see a blurry description of the designs of ONE type of seeding machine (coil) and some control principles. I believe that the name should be clarified to a specific type of seeding device or expand the article by comparing studies of the seeding rate control system for several types (for example, coil, disk, pneumatic, etc.)

The main design features of the monitoring system (L28-67) are described rather vaguely). It is necessary to clearly define the structure of the seeding rate management system in the introduction: what components does it consist of? It is desirable to provide a drawing (diagram) so that the reader can understand which component of the control system was changed.

The authors point out L79-80 that they previously developed a seeding rate sensor (that is, the design!), and I quote, "but it was not integrated into any system for monitoring the seeding rate." Why? What does this fact give the reader of the paper? Is this self-citation appropriate?

The authors indicate L84-85 (instead of goals) that "comparative experiments were conducted between the proposed approach and the traditional approach to controlling the speed of rotation of rollers". However, there is no clear definition of what the proposed approach is and what the traditional approach is. Specify clearly what is meant by the term "Approach". You may need to move some paragraphs from the M&m section.

At the end of the Introduction section, it is necessary to clearly indicate what is still being studied in the paper: approaches, sensor designs, or algorithms for controlling the process of regulating the speed of the coil? Please state your goals and objectives clearly.

In their conclusions, the authors point out (L514-516): "the Main success of this study was the DEVELOPMENT OF a SYSTEM FOR monitoring the SEEDING RATE, which can provide feedback and directly control the seeding rate." However, in the introduction, they indicate a comparison of approaches.

There is no DOI in the list of references. Give it for all articles that have it.

My main question to the authors is: what should the reader compare? An approach (a new mathematical expression used for use in the algorithm), an algorithm (a new sequence of actions for controlling the seeding rate with feedback), or a new design of the seeding rate control system? Please specify.

Please also find some suggestions for correcting the article in the attached PDF file.

Round 2

Reviewer 2 Report

The authors responded to the comments in detail, excluding any questions that arose during the review process.

I hope that the changes made by the authors will significantly increase the reader's interest.

I thank the distinguished authors and the editorial team for the opportunity to review the paper.

I believe that the manuscript can be published in its current form.

Author Response

The reviewer has agreed that the manuscript can be published in its current version. Therefore, in this round, we only need to response the academic editors' comments.